REGISTERED REPORT PROTOCOL

# Determinants of non-adherence to home injury prevention practice among parents of under-five children in North Seberang Perai district, Penang: A mixed-methods study protocol

**Nurul Iman Abdul Rahim[1], Hayati Kadir Shahar[1,2]\*, Nor Afiah Mohd Zulkefli[1], Ahmad Iqmer Nashriq Mohd Nazan[1]**

**1** Department of Community Health, Faculty of Medicine and Health Sciences, Universiti Putra Malaysia, Serdang, Selangor, Malaysia, **2** Malaysian Research Institute of Ageing (MyAgeing ™), Universiti Putra Malaysia, UPM Serdang, Selangor, Malaysia

\* hayatik@upm.edu.my

## Abstract

### Background

Unintentional injury is a global burden that occurs everywhere, including in our homes. Young children are most vulnerable to home injuries because they still develop their physical and psychological skills and spend most of their time at home. Despite being largely preventable, three consecutive national surveys show no reduction in the rate of injury among children under five. More surprisingly, children from high-income families were found to have the highest incidence of injury, contradicting the findings from other countries.

### Objective

This study aims to identify the determinants of non-adherence to unintentional home injury prevention practice among parents of under-five children in the North Seberang Perai district, Penang.

### Methods

This sequential explanatory mixed-methods study consists of two phases consisting of a quantitative study which looks into respondents and their child's sociodemographic status, their home injury prevention practice and the independent variables, followed by a qualitative study that interviews parents with non-adherence to home injury prevention practice and explore their barriers. In phase I, the parent or primary caregiver of a child age less than five years old who age 18 or older and is a Malaysian will be included in the study while being disabled or having a severe psychiatric disorder or having the index child diagnosed with chronic disease will make them not eligible to participate in the study. Derived using the two-group proportion formula, a sample size of 453 parents will be sampled among those

**Data Availability Statement:** All relevant data from this study will be made available upon study completion.

**Funding:** HKS had been awarded with the Fundamental Research Grants Scheme under the Ministry of Higher Education with code FRGS/1/2020/SKK05/UPM/02/2. The funding website can be find through this link: http://www.mygrants.gov.my/csp/sys/bi/%25cspapp.bi.work.mygrant.custom.login.cls?$NAMESPACE=MYGRANT. The funder has no influence in study design, data collection and analysis, decision to publish, or preparation of the manuscript.

**Competing interests:** The authors have declared that no competing interests exist.

with under-five children following up at the Maternal Child Health Department in the health clinics of North Seberang Perai using stratified systematic sampling. Chi-square/Fisher Exact test, simple logistic regression and multiple logistic regression will be used for data analysis. The sample will be stratified according to household income to look for associated factors and determinants of low prevention practice. In phase II, parents with a low score from the quantitative study will be selected to participate in the qualitative study using purposive sampling. A semi-structured interview using the help of an interview guide will be carried out and recorded with a voice recorder. The thematic analysis approach will be used to analyse the qualitative data.

## Results

The study has been registered under the National Medical Research Registry.

## Conclusion

It is hoped that findings from this study can shed light on the barriers faced by under-five parents in carrying out preventive measures at home.

## Introduction

Unintentional injury is a global public health burden. It was estimated that in the year 2013, 973 million people sustained injuries that required medical attention, where 4.8 million resulted in death [1]. UNICEF describes the burden of injury as similar to the iceberg phenomenon refers to a situation where what we see is just a small portion of the actual size of the problem. In this case, most child home injuries are hidden from the public because they are unreported. For every single death due to injury, there were about 160 hospital admission and more than 2000 emergency department visits [2]. The burden of injury is almost two times higher than that of Tuberculosis and Human Immunodeficiency Virus. However, it was never addressed as a priority agenda for most countries, including Malaysia [3]. In Malaysia, injury mortality for children under the age of 5 years remained unchanged from the years 1990 to 2013 inclusive, despite a marked reduction in death from all communicable diseases [4]. Further, 8% of all mortality in children under the age of 5 years is caused by injury, with drowning, followed by road traffic collisions, burns, choking and poisonings as the leading causes of death [5].

For a child between the age of one and four, injury is the leading cause of death [6]. Because they spend most of their time at home, the majority of the injuries they sustain occur where they grow up [7]. Therefore, it is important to provide a safe condition for children's learning and exploration when they are incapable of recognising hazards and risks [8]. Childhood injury does not only cause health and social burden to a child and his or her family, but it was estimated to cost around USD516,938 to a staggering USD9,550,704 per year [9].

Child injury prevention is the responsibility of many parties starting from parents/guardians, community and private and government organisations either locally or globally. For children under-five who stay at home for most of the day, parents or caregivers play a large role in ensuring the safety and security of their children. Rezapur and colleagues have pointed out that the prevention of home-related injuries is very important to prevent morbidity and

mortality [7]. Injury incidence at home was found to be significantly associated with poor parental prevention practices on their child's safety [10, 11].

Globally, the burden of injury is heaviest among the poor, especially among children who come from low-income families [12]. Low income was strongly associated with most causes of injury mortality, particularly fire/burn and poisoning [13]. Various studies from low, middle and high-income countries have shown a significant relationship between poverty and child injury [14–16]. The studies explained two groups of factors; parent factors such as parental depression and risky behaviours and community factors such as substandard safety measures in their low-income house, decreased access to safe recreational activities and high crime rate [17, 18]. However, the case is not the same in Malaysia. Based on the National Health and Morbidity Survey 2006, 2011 and 2016, the prevalence of child injury is highest in a high-income families [19–21]. The contradicting finding is unique to Malaysia, and to our knowledge, has never been explored yet.

To date, there is no known mixed-methods study on injury prevention in Malaysia. Most studies are quantitative and focus on injury rate rather than its prevention. It is important to explore the issue from the perspective of the local context to understand the reality of unintentional home injury prevention practice in Malaysia. Without an understanding of the lives and lay expertise of those on the receiving end of our well-meaning efforts, we risk ineffective, or worse, intrusive and harmful interventions, which may, for instance, raise the level of anxiety about risk while doing nothing to reduce the risk itself [22].

## Moronggiello and corbett model

Morrongiello & Corbett (2008) proposed a conceptual model (Fig 1) that shows the interplay between three important factors; parent, child and environment and their influence on parents' decision to carry out injury prevention. Each component gives weight to the parent's discretion and results in the parent's behaviour. This model also considers macro-level factors such as economic status and cultural norms [23].

## Conceptual framework

This conceptual study framework is shown in Fig 2. The independent variables are formed by the model in Fig 1 which shows three groups of variables; 1) parent characteristics, 2) child characteristics and 3) environment characteristics that contributed to parental home injury prevention practice. The parent characteristics combined the constructs of the Health Belief Model (HBM) and one construct from the Theory of Reasoned Action which is the subjective norm or the perceived social norm, and other single characteristics such as personality and parenting type. The child characteristics were made of child age, sex, behavioural attributes, temperament and injury history. The environmental characteristics include the type and number of hazards. The model also acknowledges the influence of macro-level factors such as the economy, local culture and ethnic norms.

The theories (HBM and TRA) that have been adapted into the model are well known to be able to explain injury prevention practices as proven by previous studies [24, 25]. Although this model is not well tested yet, it is the only known model built specifically on child injury prevention subjects and the dynamic interplay between multiple factors shows a holistic perspective of the model, making it the best choice for our conceptual framework. The outcome of the model that states parents' decision for prevention can go both ways either to carry out or not to carry out the practice, which we believe may be able to explain this study outcome which is non-adherence to home injury practice.

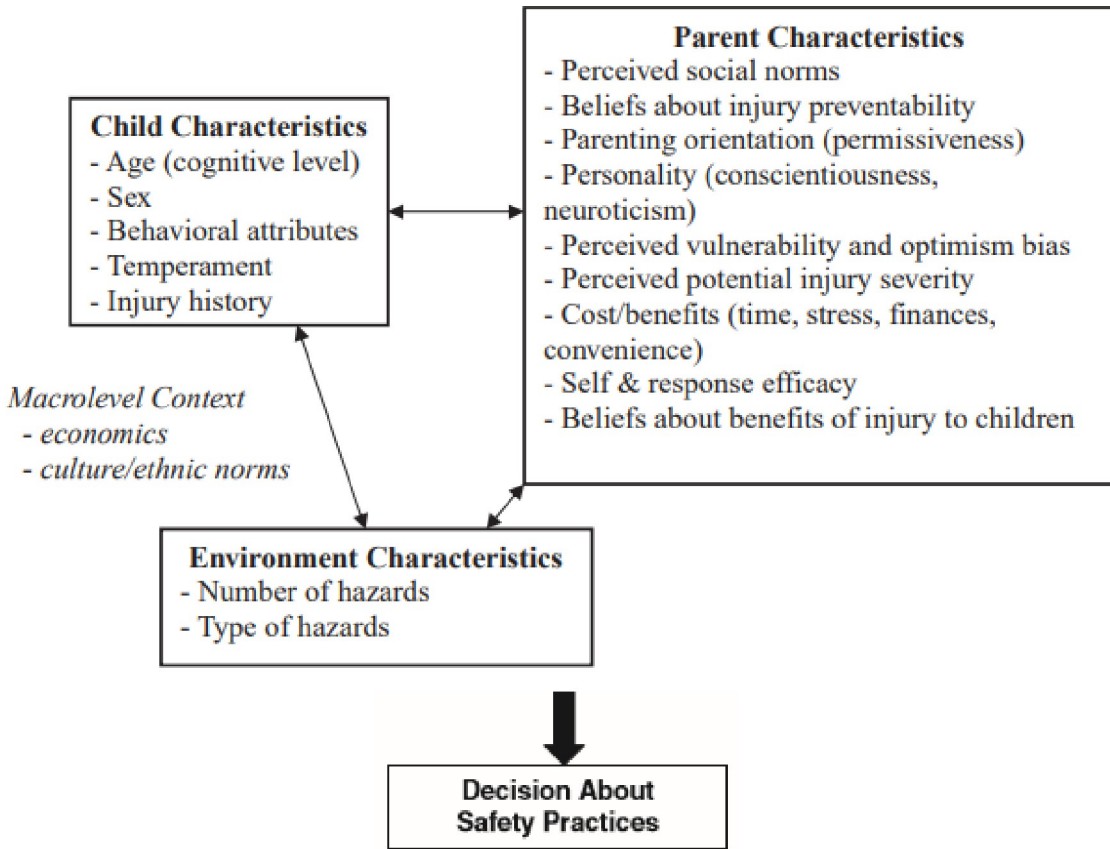

**Fig 1. Conceptual model representing the dynamic interplay of factors that affect caregivers' decisions about whether to implement safety precautions to prevent injury to young children.** Source: Morrongiello & Corbett (2008).

### Study objectives

Quantitative research objectives:

i. To determine the prevalence and type of unintentional home injury among under-five children;

ii. To determine the association between the independent variables with low unintentional home injury prevention practice among parents of under-five children in North Seberang Perai District, Penang.

iii. To determine the determinants of low unintentional home injury prevention practices among parents of under-five children;

Qualitative research objective:
To explore the contributing factors of low unintentional home injury prevention practice level among parents of under-five children in North Seberang Perai district, Penang.
Mixed methods objective:
To explain the parents' non-adherence to unintentional home injury prevention practice (quantitative study) with the findings of the qualitative study.

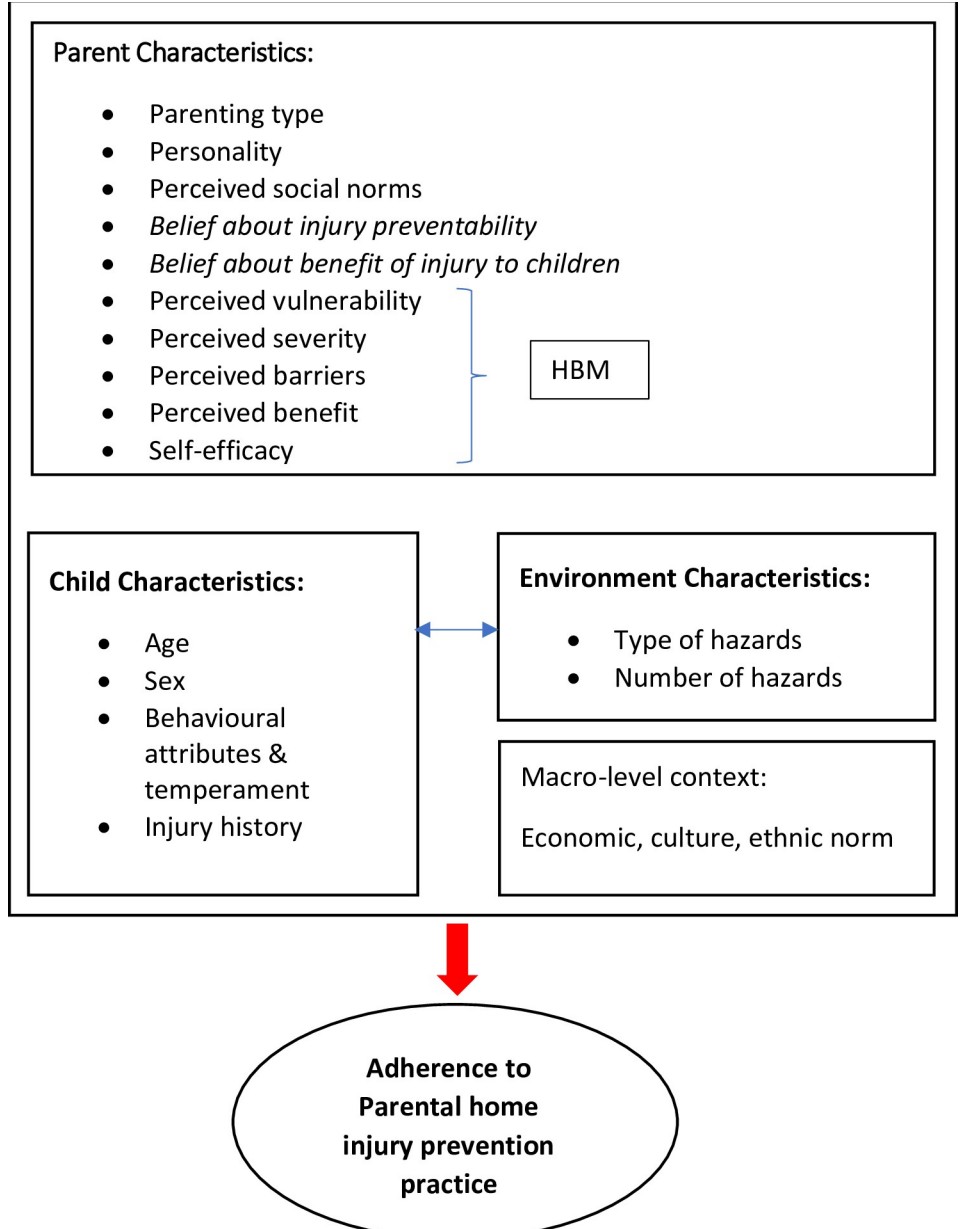

**Fig 2. Conceptual framework of the study.**

## Methods

### Ethics approval

This study has been approved by the Medical Research and Ethics Committee on 7th of October 2020 with ID# NMRR-20-1982-55320 (IIR).

### Mixed methods design

Mixed methods study is "an approach to research in the social, behavioural, and health sciences in which the investigator gathers both quantitative (closed-ended) and qualitative data

(open-ended), integrates the two, and then draws interpretations based on the combined strengths of both sets of data to understand research problems" [23]. In this explanatory sequential mixed methods design, the integration of the qualitative and the quantitative data occurs twice; first during the first phase where the quantitative results help plan the follow-up qualitative data collection and the second integration when the results of the quantitative study are connected with the results of the qualitative study to derive a conclusion whether the qualitative results can explain the quantitative results.

This integration of qualitative and quantitative data will occur when the quantitative results are used as the inclusion criteria of qualitative participants and act as a guide in developing the interview guide. The second data integration will occur when qualitative data can explain the quantitative results. For example, if parental knowledge is a significant variable (in quantitative research), the interview during qualitative research should focus on how much you ever heard or learnt about home injury prevention.

## Quantitative study (Phase I)

This quantitative research uses an analytical cross-sectional (observation) design where the data is collected at one point in time without introducing any intervention to the participants. This cross-sectional analytical study will be carried out in North Seberang Perai district, Penang from 1st November 2020 until 31st January 2021. The study populations are parents of children aged less than five years old who has a follow-up in any Maternal Child Health department in any health clinic in North Seberang Perai, Penang. The study site (Maternal Health Clinic) will be chosen as the clinic could provide a list of parents of under five-aged children for sampling purposes. It is not feasible to get a sampling frame from a community-based registry of parents with under-five children.

There are several inclusion criteria to look for when recruiting respondents for the study. The inclusion criteria are:

i. The parent or primary caregiver of a child aged less than five years old. If there is more than one child aged below five, the younger will be chosen as the index child; This is because children below five years old are at the most risk of sustaining fatal home injury. The youngest of the siblings is chosen as studies have shown that parents will be focusing and carrying out most prevention on the youngest as they would think they are the most susceptible [26, 27].

ii. Aged 18 and above, adults are chosen as they are considered well-developed mentally, physically and socially to take care of children.

iii. Malaysian citizens who can provide his/her valid identification card. We only choose Malaysian citizens because the income and social gap between Malaysians and non-Malaysians are quite big which may influence many factors including knowledge and social norm. This might cause an outlier in the study findings later on.

The exclusion criteria of the study are:

i. Parents/caregivers of a child who has an underlying chronic disease, including learning disorder. It is acquired from a doctor's documented diagnosis from a medical report or in the child's record book;

ii. Parent/primary caregiver has an underlying severe psychiatric disease. It is acquired from a medical report or psychiatric clinic follow-up record book/card, or;

iii. Parent/primary caregiver has a permanent disability. It is acquired either from a medical report or a disability card.

The sample size for the study was calculated using two group proportions formula by Hosmer & Lemeshaw (1990) [28]. Adjusting for 20% dropout, the sample size for this study is estimated to be 453.

Prior data by Honda & Nagata [29] that look into the associated factors of injury prevention practice indicated that the proportion of parents with poor adherence to injury prevention practice with index male child is 0.416 and the proportion of the parents with poor adherence to injury prevention practice with female index child is 0.276 [29].

$$n = \frac{\left(z_{1-\alpha}\sqrt{2\bar{P}(1-\bar{p})} + z_{1-\beta}\sqrt{P_1(1-P_1) + P_2(1-P_2)}\right)^2}{(p_1 - P_2)^2}$$

n = sample size estimation

Z(1-$\alpha$) = Standard error associated with 95% confidence interval = 1.96

Z(1-$\beta$) = Standard error associated with 80% power = 0.842

$P_1$ = population proportion 1

$P_2$ = population proportion 2

$\acute{P}$ = $(P_1+P_2)/2$

$P_1$ = 0.416 (Proportion of poor adherence parents with male index child)

$P_2$ = 0.276 (Proportion of poor adherence parents with female index child)

$\acute{P}$ = $(P_1+P_2)/2$ = (0.416 + 0.276) / 2 = 0.346

$$n = \frac{\left\{1.96\sqrt{2(0.346)[1-(0.346)]} + \left[0.842\sqrt{(0.416)(1-0.416) + (0.276)(1-0.276)}\right]\right\}^2}{(0.416 - 0.276)^2}$$

n = 181.21

The minimum sample size per group is 181.

There will be two groups of children age in this study, therefore 181 x 2 = 362

For this study, 20% drop out is used for sample size adjustment.

Adjust for 20% drop out: 362 / 80% = 452.5 → rounded to 453.

Therefore, the sample size for this study is 453.

Stratified systematic sampling method will be used to sample the study population. In the North Seberang Perai district, there are seven health clinics which are Butterworth Health Clinic, Kepala Batas Health Clinic, Penaga Health Clinic, Sungai Dua Health Clinic, Mak Mandin Health Clinic, Tasik Gelugor Health Clinic and Kuala Muda Health Clinic. The number of samples taken from each clinic is stratified according to the Maternal Child Health workload of under-five patients per clinic in the past year. Table 1 shows each clinic's under-five workload for the year 2019.

For each health clinic, the list of under five under follow-up for the day will be used. Every 3rd patient (a random number generator generates the number 3) from the list will be selected.

**Table 1. The under-five patient workload in PKDSPU clinics in 2019.**

| No | Health clinic | Number of under-five attendance | Percentage (%) | Number of samples |
|----|---------------|--------------------------------|----------------|-------------------|
| 1 | Butterworth | 11,784 | 12.7 | 58 |
| 2 | Mak Mandin | 13,294 | 14.3 | 65 |
| 3 | Kepala Batas | 24,215 | 26.0 | 118 |
| 4 | Tasik Gelugor | 14,363 | 15.4 | 70 |
| 5 | Penaga | 13,821 | 14.5 | 66 |
| 6 | Sungai Dua | 13,689 | 14.5 | 66 |
| 7 | Kuala Muda | 1,865 | 2.0 | 10 |
| | **Total** | **93,031** | **100** | **453** |

We use systematic sampling method where individuals are selected at regular intervals from the sampling frame. The sampling frame will be an appointment list where not only under-five children but all patients under The Maternal Child Health unit will be listed too. From the daily list, we will exclude the non-under-five patients, and then from the list, we will choose every third to be approached. Number 3 is chosen because during the pandemic time, the number of patients reduced markedly so if we choose a bigger number, we might not have enough patients for a day of data collection as the clinic only runs from 8 am to 5 pm on week-days. The parent will be checked for eligibility and counselled for consent. The questionnaire is to be completed by her or himself.

**Data collection.** Data collection will be carried out by the researcher and enumerators. Before data collection is carried out, all enumerators will be trained on how to select the respondents according to the inclusion and exclusion criteria. They also will be taught on how to give informed consent and how to use the questionnaire based on the child's age. In the case of both parents being available, the parent who spent the most time with the index child will be chosen as the respondent. Once the parent is selected, they will be given a briefing on a few instructions: what the study is all about, what they are supposed to do if agree to participate, what are the benefit and risks they might receive by joining the study and the freedom to with-draw from the study at any time during the data collection process.

**Study instrument.** There will be nine sections in the questionnaire that cover the three factors in the model; parent, child and environment. The sections are A. Sociodemographic status of parent and child, B. Child temperament, C. Parenting style, D. Parent perception (Health Belief Model), E. Social norm, F. Parent's knowledge, G. Home injury prevention practice and I. House hazards.

*Section A*: *Sociodemographic status of parent and child.* This 13-items section will cover the parent's age, sex, ethnicity, education level, income level, occupation, marital status and type of house, while for the child, there will be age, sex, number of siblings, injury history and type of injury.

*Section B*: *Child temperament.* The child temperament assesses aspects of the child's usual behaviour, including activity level, affective attributes, attachment styles, compliance, and sociability. The temperament scales were adapted from Rothbart's Infant Behaviour Questionnaire and Kagan's compliance scale. The questionnaire will be measured by a set of age-appropriate questions. For a child aged less than 12 months, 42 questions need to be answered, for a child aged 12 to 23 months, the parent needs to answer 32 items and for the child aged two to four, 13 questions need to be answered. Reverse scoring means that the numerical scoring scale runs in the opposite direction. The behavioural tendencies of the child are rated by the parent on a seven-point scale, ranging from 'Never' (value of 1) to 'Always' (value of 7). So, in

reverse scoring, 'Never' would give a score of 7, 'Very Rarely' would be 6, 'Less than half the time' equals 5, 'About half the time' stills 4, 'More than half the time' equals to 3, 'Almost always' equals to 2, 'Always' will be 1. The scores of the various scales are computed by summing the individual items in the scale. Some items are recorded in reverse before summing. The question with reverse-coded items is in bold to ease the scoring process. If any item component of a subscale was missing, that score would not be computed.

*Section C*: *Parenting style*. This 30-item section was adapted from Nor & Sutan (2020) [10]. The questionnaire provides three different types of parenting including laxness, reactivity and hostility. Laxness refers to a parent's inconsistent and permissive parenting while over–reactivity refers to a parent's harsh or punitive parenting. Hostile parenting refers to the extent to which a parent hits, curses and insults their child. The higher score indicates a dysfunctional parenting style.

*Section D*: *Parent perception (Health Belief Model)*. This section will have 56 items which consist of 11 items for perceived vulnerability, 11 items for perceived severity, 12 items for the perceived barrier, nine items for the perceived benefit and 13 items for self-efficacy. The instrument has been developed, validated and checked for construct correlation by Russel (1991) [30].

*Section E*: *Perceived social norm*. A five-point Likert scale ranging from "not at all" to "very much" will be used to see how six individuals or parties influence the respondent's decision on carrying out home injury prevention practice. These people are: 1) partner/boyfriend/husband, 2) close relative, 3) close friends, 4) nurse or doctor, 5) social worker or caseworker, and 6. minister or religious leader. The items have been validated with good reliability by Russel (1991) [31].

*Section F*: *Parent's knowledge*. This questionnaire is adapted from Nadeeya et al. (2016) who carried out a study among Malaysian parents [32]. There were 16 items measuring mothers' level of knowledge which includes knowledge related to hazards and injuries that could happen to children at home. Question 1, 4, 7, 8, 12, 13, 14, and 16 with agree answers and 2, 3, 5, 6, 9, 10, 11 and 15 with disagree answers, that the correct answer is given a score of 1 while the wrong and unsure answers will be given a score of 0. The total knowledge score can range from 0 to 16. It has been validated by the mentioned author and found reliable with an acceptable Cronbach Alpha value of 0.602.

*Section G*: *Home injury prevention practice*. The 21-item questionnaire was adapted from Mohamad Nor & Sutan (2020) as it was validated and culturally adapted for the Malaysian population. A question which is correctly answered will be given one mark while a wrongly answered will be given 0 marks [31].

*Section H. House hazards*. This section has 27 common hazards at home that parents need to note as present (1 mark) or not (0 mark). It was adapted from Tertinger, Greene & Lutzer (1984) [32].

Face validity will be carried out with five parents from the same population who are not included in the study. One parent will be approached and all of their incomprehension or confusion will be addressed until they can fully understand the whole questionnaire. They will provide feedback on their understanding of the items in the questionnaire. Corrections will be done in response to the comment, and the process repeats until all items are cleared. This process will be repeated with multiple parents (one at a time) until there is no more issue is raised.

Meanwhile, for content validity, subject matter experts (SMEs) from different fields will be asked to review the contents of the questionnaires. For content validity, we plan to approach

eight subject matter experts (SMEs) from various expertise. They include a public health specialist, a family medicine specialist, an injury specialist and a paediatrician. The feedback will be in the form of whether the item is essential, useful, but not essential or not necessary for the research. They will be looking into the clarity, accuracy, language and cultural relevance of every item. The feedback will be used to measure the content validity ratio (CVR) with the calculation described by Ayre & Scally (2014) [33]. They will be given the questionnaire, and at the end of each item, there is a box for them to rate as 1 to 3, 1: not necessary, 2: useful but not essential and 3: essential. Next to the box is another box for them to write their comment if any. For each item, CVR is calculated with the formula CVR = (n—N/2)/(N/2), where n is the number of SMEs who describe an item as essential and N is the total number of SMEs involved. Only items with a CVR of more than 0.9 will be accepted.

The reliability of this questionnaire will be examined using the internal consistency method. Once ethical approval is received, a pre-test will be distributed to a group of respondents that were not included in the sample size but came from the study population. The study will test the reliability number of individuals based on 10% of the sample size as pre-test respondents following Connelly's (2008) [34] suggestion. Good reliability is achieved with a Cronbach Alpha level equal to or greater than 0.7 [35]. The original questionnaire is developed in English, and it will be translated into the Malay language by an expert Malay-English translator and then back-translated into the English language by another expert Malay-English translator. This study is awaiting ethical approval from the Medical Research and Ethics Committee (MREC).

**Data analysis.** Data entry and statistical analysis will be carried out using IBM SPSS version 26.0. Continuous variables will be reported using the mean (standard deviation) or median (interquartile range) and checked for normality. Meanwhile, for categorical data, the prevalence was presented using frequency and percentage. The associated factors of non-adherence to home injury prevention practice will be checked using Chi-Square or Fisher Exact test. Simple logistic regression will be used to look for a significant relationship between the variables, and multiple logistic regression will be used to examine the predictors of parental non-adherence behaviour. Significance is achieved with a p-value of less than 0.05. The data will be presented according to the level of analysis and in a separate table: Parental characteristics, child characteristics and environmental characteristics. Firstly, the descriptive data will be presented in frequency and percentage. Then, Chi-Square/Fisher Exact test results will present the n, %, $X^2$ and p-values according to categories in each variable. Lastly, a table of multiple logistic regression will be presented with Beta coefficient, standard error, odds ratio, 95% confidence interval and p-value will be presented.

## A qualitative study (Phase II)

A qualitative case study will be used for the qualitative phase where the parents' experience and perspective will be used to provide insight into their low level of unintentional home injury prevention practice and its contributing factors. A case study facilitates exploration and comprehension of something unexplored and unique such as parents' perception of child home injury prevention [36].

**Data collection method and process.** This study will be carried out in North Seberang Perai district, Penang from February until July 2021. A field visit is an essential step in the data collection process. It is to ensure that the researcher is familiar with the research setting and environment before data collection. The interview can be carried out at the clinic or in the respondent's house. Before data collection, the researcher needs to familiarise herself with the

respondent by conducting a preliminary home visit. It is to help build trust and rapport between the researcher and respondents, and in turn, smoothen the research process.

Purposive sampling will be employed to select the respondents in the qualitative phase. Purposive sampling is stopped when the saturation point is reached, where there is no more new information gained after multiple interviews to form themes that answer the research questions when there is enough information to replicate the study, when the ability to obtain additional new information has been attained, and when further coding is no longer feasible [37]. The sampling population in the qualitative study (Phase II) should fulfil the following criteria:

i.  Had score less than the mean score for Home Injury Prevention Practice Questionnaire

ii.  How about significant predictors from your Quantitative study?

Participants of different ages, gender, race, marital status, education and income level will be included to achieve maximum variation. The potential respondents will be given verbal and written descriptions of the contents of the interview.

A semi-structured interview will be used in this study, as it is considered to be the most suitable method [38, 39]. The core advantage of semi-structured interviews is that informants have the chance to voice out their opinions without fear, whereas the interviewer could still control the direction of the interview.

The interview will be conducted with the help of an interview guide. It contains a set of leading questions and probing questions according to the research question to get a better and deeper understanding of the respondents' explanations of their adherence to injury prevention practices [40]. The interview guide is prepared with the guidance of a few qualitative experts and some literature. A pilot study will be conducted on two parents of under five, who also had a low score of adherence to injury prevention practice to assist in the improvement of the interview guide. The purpose of the Pilot Test is to evaluate the feasibility of sample size, time, cost, risk, and performance of our research project. The pilot study will be carried out with the interview guide prepared and the researcher will note any issues developed during the interview and address them by making changes to the interview setting, interview guide, ways of conveying questions and responses etc. When no more issue rises during the interview, the actual study can start.

Before an interview, respondents will be briefed that their involvement is entirely voluntary, and all the information derived from the interview will be kept confidential. They are also will be informed that they may choose not to answer any question if they feel uncomfortable, and they have the right to interrupt the interview at any point in time. Permission to record the interview will also be obtained from the respondent. Verbal and written consent will be obtained to ensure the respondent's confidentiality before the interview. Each interview will be started with an ice-breaking session to make sure that both the interviewer and the interviewee are comfortable with each other. The interview will begin with general conversation such as sociodemographic information, daily activities and introduction to the interviewee's child (or children). It is followed by asking the questions in the interview guide (S1 Appendix).

The focus of the interview is to explore the parents' perception and attitude towards child home injury prevention. The questions are open-ended (e.g., what do you think of injury, why do you partake in a safety measure?) and then, will be followed up with additional probing questions to the main questions (e.g., can you elaborate further or can you explain to me with some examples?). Although the interview guide is available, the conversation must not only limit the questions listed. The researcher needs to welcome new thoughts and should not impose her own preconceived opinions about the study.

**Validity of the interview.** In qualitative research, the data needs to be rigour which reflects trustworthiness and truthfulness, this is based on the researcher's credibility, transferability, dependability and confirmability [41]. Credibility assures those research findings represent the credible information gathered from the novel data from the interviewees and provide a real interpretation of the interviewees' real views. For this study, credibility will be ensured through the 'member check' process [41]. Once the transcripts of the interview are ready, they will be read to the interviewee to check whether they agree with the content. The correction will be made based on the interviewee's comment to avoid false interpretation. Peer review also needs to be done to check whether the findings suit the qualitative data.

Transferability of this study is ensured by detailing all the research processes from study location, sampling population, sampling methods, interview questions and so on to help other researchers to evaluate whether the study is applicable in another setting.

Similar to reliability for a quantitative study, dependability shows the extent to which the research would produce similar findings if it is to be repeated. To ensure this quality, this study provides detailed methodological information of the study, which will help other researchers to carry out a similar study in their setting. Audit trail; a detailed explanation of the data collection and data analysis process will be prepared.

The confirmability of this study will be ensured by taking detailed field notes during the interview. After the interview was transcribed, the researcher needed to listen to the recording to verify the transcription's accuracy.

**Sample size.** In a qualitative study, there is no formula or pre-determined sample size, like in a quantitative study. There is great variability in the qualitative sample size, but most of the references in the journal articles and book chapters recommended by any number between five to 50 respondents as sufficient [42]. In this study, qualitative data collection was carried out until it reached a saturation point; no new information was obtained from the respondents [43].

**Data management.** Data management is vital to make sure that the study findings were analysed in an orderly and accessible fashion. An individual folder will be created for each participant containing their socio-demographic information, including regarding their child and their home. Apart from that, the date, time and duration of the interview will be written on the transcript. NVivo Version 12 software will be used for this purpose and also for data analysis where the data will be coded according to the wordings or phrases in the transcription. A codebook will be used to document recurring words or meanings that would later be coded and grouped into categories or themes.

**Data analysis procedures.** The interview content will be analysed while transcribing the interview audio. The initial findings obtained by the researcher will provide a hint and probe for a further interview with the next interviewee. It allows a high quality and detailed description of qualitative data. Audio recordings will be transcribed by listening to the interview repeatedly and writing each word and sound made by the interviewee. The Malay language will be used to interpret the data. No translation will be done for the verbatim to maintain its originality and to prevent misinterpretation during analysis.

Thematic analysis will be carried out by two researchers who are experienced in the thematic analysis process. The steps consist of familiarization, data coding, identifying themes, reviewing themes, refining themes and producing a report. At first, familiarization will be done by re-reading each transcript to be immersed and acquainted with the data. The second step involves initial data coding to identify the key features of the interview. The interview text will be organized by labelling different words or phrases or sentences into certain categories. Next, we will identify the themes by combining the codes into themes and scrutinizing the codes to identify significant ones, but broader patterns of themes. The fourth step is to review the themes by comparing the themes that emerged in each transcript to determine whether

these data could answer the research questions. In this regard, the themes should be meaningfully connected to the codes identified earlier and should be differentiable. The next step is to refine the themes, which involves defining, naming, and finalizing the themes. This process includes a comprehensive analysis of each theme which focuses on the deeper meaning behind each theme. The last step is to produce the report where all the analysis will be used with supporting verbatim for each theme. The report will consist of a result of analysis of the verbatim, themes that come out from the analysis and the discussion of the themes. An unexpected result that does not belong to any theme may also be included in the report.

**Ensuring rigour.** To ensure rigour in the qualitative data gathered from the interview, we need to make sure credibility, transferability, dependability and confirmability are in place [44].

For this study, credibility will be ensured through the 'member check' process in which after the transcripts of the interview are ready; they will be read to the participant interviewed to check whether they agree with the content. The correction will be carried out based on the participant's comment. It helps to address researcher bias and overcome any wrong description [45]. Peer review also needs to be done to check whether the findings are authentic. For this, two qualitative experts will be approached for assistance.

Transferability of this study is ensured by detailing all the research processes from study location, sampling population, sampling methods, and interview questions to help other researchers to evaluate whether the study is applicable in another setting.

This study will ensure dependability by providing detailed methodological information of the study, which will help other researchers to carry out a similar study in their setting. Audit trail; a detailed explanation of the data collection and data analysis process will be prepared.

Confirmability of this study will be ensured by taking detailed field notes during the interview. After the interview was transcribed, the researcher needs to listen to the recording to verify the transcription's accuracy. Field notes also can be helpful during the data analysis process to understand what the interviewee meant from their description.

## Results

The study has been registered with the National Medical Research Register (NMRR). Data collection is expected to start around November 2020.

## Discussion

### Contribution to the literature

This study will determine the under-five parents' adherence to home injury prevention practice and its contributing factors from a quantitative and qualitative perspective. The in-depth exploration of parents' reasons for non-adherence is the first-ever in Malaysia and may bring new details for use in further research. This study will also determine whether there are distinct contributing factors between different income levels and if present, may be explained by the qualitative data from the interviews with the parents.

### Challenges

The challenge that might be faced during this study is the COVID-19 pandemic situation where the researcher needs to meet a lot of participants and spends a lot of time with them especially for interviews, making the researcher at risk of contracting COVID-19. Therefore, it is important to wear PPE such as facemask and face shield and observe good SOP at all times. The clinics will have reduced patients as they do need to reduce the crowd. Some clinics even

reduce the appointment to only half day. This will cause a lack of patients to be approached in a day, which may prolong the time to achieve an adequate sample size. Therefore, we had to sample a bit more frequent where every third from the sampling frame in order to complete the study in the planned time. Moreover, health clinic-based studies can introduce some bias including selection bias as there is certain demography that is most likely to attend government health clinics. This will be addressed by doing an analysis with a standardized level of income. Other than that, more previous studies only related to injury rates than their prevention. This is challenging for the researcher to find references for designing the study. Last but not least, sequential mixed method design will take a longer time to be completed. The second phase might be delayed if there is hiccup faced in the first phase. However, the large sample size will be collected by only a small team (principal researcher and two enumerators) within a limited study duration.

## Limitation

For the quantitative study, the limitation is in the design where it is a cross-sectional study that reports data from one point in time and is unable to show the temporality. The majority of the data of this study will be self-reported by the respondents, which depends heavily on the respondent's honesty. Presuming the limited availability of respondents, the hazard data will also be collected from self-reported data instead of direct observation by the researchers. Finally, the quantitative part of the study will not be measuring external factors such as economic status and ethnic norms due to limited instruments. Another limitation that arrives with the pandemic is during the in-depth interview process. As participants need to wear masks during the interview, it will be a challenge to observe their facial expressions which is vital in qualitative data analysis. This limitation will hopefully be supported by participants' intonation and body language which can still be visible during the interview.

## Conclusion

This study is expected to be able to explain the non-adherence of home unintentional injury prevention practice among parents of under-five with in-depth qualitative results and to find a significant association between the dependent variable and the independent variables. We also hope to be able to determine the predictors of low unintentional home injury prevention practice scores.

## Supporting information

**S1 Appendix. Interview guide.**
(DOCX)

**S1 File. Questionnaire–English version.**
(DOCX)

## Acknowledgments

No copyrights were violated in the process of the research and manuscript writing.

## Author Contributions

**Conceptualization:** Nurul Iman Abdul Rahim, Hayati Kadir Shahar.

**Methodology:** Nurul Iman Abdul Rahim, Hayati Kadir Shahar, Nor Afiah Mohd Zulkefli, Ahmad Iqmer Nashriq Mohd Nazan.

**Writing – original draft:** Nurul Iman Abdul Rahim.

**Writing – review & editing:** Nurul Iman Abdul Rahim, Hayati Kadir Shahar,
Nor Afiah Mohd Zulkefli, Ahmad Iqmer Nashriq Mohd Nazan.

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
