## [Decision Letter · Decision Letter 0]

1 Dec 2021

PONE-D-20-40899Determinants of non-adherence to home injury prevention practice among parents of under-five children in North Seberang Perai district, Penang: A mixed methods study protocolPLOS ONE

Dear Dr. Kadir Shahar,

Thank you for submitting your manuscript to PLOS ONE. After careful consideration, we feel that it has merit but does not fully meet PLOS ONE’s publication criteria as it currently stands. Therefore, we invite you to submit a revised version of the manuscript that addresses the points raised during the review process.

Your study protocol has been assessed by three reviewers and their comments are appended below. They raised important, constructive comments regarding the methodological and presentational aspects of your work. Please pay particular attention the comments related to the thorough reporting of data analysis, as these are critical for the publication of a study protocol. Given the extensive comments raised, it is likely that your submission will undergo a second round of peer-review once your revisions have been submitted. 

We look forward to receiving your revised manuscript.

Kind regards,

Dario Ummarino, Ph.D.

Senior Editor

PLOS ONE

Journal Requirements:

No authors have competing interests.

Reviewers' comments:

Reviewer's Responses to Questions

**Comments to the Author**

1. Does the manuscript provide a valid rationale for the proposed study, with clearly identified and justified research questions?

Reviewer #1: Yes

Reviewer #2: Partly

Reviewer #3: Yes

2. Is the protocol technically sound and planned in a manner that will lead to a meaningful outcome and allow testing the stated hypotheses?

Reviewer #1: Yes

Reviewer #2: Partly

Reviewer #3: No

3. Is the methodology feasible and described in sufficient detail to allow the work to be replicable?

Reviewer #1: No

Reviewer #2: No

Reviewer #3: Yes

4. Have the authors described where all data underlying the findings will be made available when the study is complete?

Reviewer #1: Yes

Reviewer #2: Yes

Reviewer #3: No

5. Is the manuscript presented in an intelligible fashion and written in standard English?

Reviewer #1: Yes

Reviewer #2: No

Reviewer #3: Yes

6. Review Comments to the Author

You may also provide optional suggestions and comments to authors that they might find helpful in planning their study.

Reviewer #1: Question 1: The rationale and justification to conduct this study are well described and understood.

Question 2: Need some improvements on the objectives arrangement. It is best to put objective 3 (the association between independent variables and low injury practice) as objective no 2 and determinants of low injury practice as no 3.

Question 3: Methodology is feasible but some part did not mention in detail. Need improvement.

Some of the comments that need improvement:

Methodology for Qualitative phase: It is best if author provides the details of Question/Probing questions that are planned to be asked in this study.

Question 4: No comment

Question 5: Some errors showed in the reference section. A typo error on Institute of Public Health reference no. 19.

References: author needs to review the standard way of writing the references following the format for PLOS. There are few references need to be looked into ie. no 6 (is this from a journal or a book/others), no 9 (is this from journal or other?), no 19, 20, 21 (IPH: standardise the name), no 10 & 26: redundant, no 31. (Capital letter for Journal's name)

Reviewer #2: PONE-D-20-40899

Overall comments: This paper requires a significant review from a sentence structure and grammatical perspective. There are several fragmented sentences and areas where words are pluralized when they should not be, or vice versa. There are several areas where the authors need to provide justification for methodological choices and areas where much more information is needed. There should also be a section that describes the challenges that may be presented in doing this work and how the authors plan to mitigate these challenges.

Specific comments:

Introduction:

The data presented in the in the introduction is outdated, please update with more recent data.

I appreciate the iceberg analogy – please explain this further in your description for readers that may not be familiar.

You should spell out TB and HIV.

You need to review your paper for tense. Some areas speak in present and others in past tense. Should be consistent.

Is there a way to have another review of this paper to revise for sentence structure? I am not sure if it is a non-English as a first language issues, but it would help readers to do another review for clarity. For example: In Malaysia, injury mortality among under-five was found to be static since 1990 until 2013 despite a marked reduction in death from the communicable disease [4]. Eight percent of under-five mortality in Malaysia is contributed by injury and the leading cause

of death is drowning followed by road traffic accident, burn, choking and poisoning [5].

This would be clearer if written this way: In Malaysia, injury mortality for children under the age of 5 years remained unchanged from the years 1990 to 2013 inclusive, despite a marked reduction in death from all communicable diseases. Further, 8% of all mortality in children under the age of 5 years is caused by injury, with drowning, followed by road traffic collisions, burns, choking and poisonings as the leading causes of death.

There are several areas where words are pluralized, and shouldn’t be.

Page 3: Injury incidence was found to be significantly associated with poor prevention practice [10].

Is there a context to this sentence? Injuries in the home specifically?

Page 4: A local study also shows similar findings where only five to eleven percent of children were found to wear their seatbelts while riding cars and only around 30 percent of children were

found wearing helmets while riding motorbikes [11].

Several issues here: First, you are making the point about injuries in the home and this study does not support that, further, children are reported to be riding motorcycles? I would imagine this is as passengers? Please clarify.

Page 4: Based on the National Health and Morbidity Survey 2006, 2011 and 2016, the prevalence of child injury is highest in a high income family [19][20][21]. The contradicting finding is unique to Malaysia, and as to our knowledge, has never been explored yet.

Please clarify what type of injury you mean here, all injury? All unintentional injury?

Page 4: Most studies are quantitative and focus on injury rate rather than its prevention.

This sentence is misleading. Quantitative studies contribute to the breadth of knowledge for prevention including, but not limited to randomised controlled trials that evaluate the effectiveness of prevention interventions. Further, given you are using a mixed methods approach, you should justify the use of both methods. What research design are you using for the “quantitative” study?

It is unclear why you chose the inclusion criteria that you have, please explain. What selection bias is inherent when you chose families that visit maternal child health clinics?

There needs to be more information on your sample size calculation.

Page 9: Every 3rd patient (a random number generator generates number 3) from the list will be selected.

This is actually not random, please explain the use of this method.

Page 11: The score will be calculated by summing the individual items in the scale. The items in bold need to be recoded in reverse before summing.

Please explain what this means.

Page 14: Your data analyses section is quite short. Please provide more explanation to how you are modeling the data, what variables you will use in your model, how the data will be presented. For the qualitative analyses, how will you present these data? By high level theme only? How will the data be categorized?

Page 15: The sampling will be stopped when the saturation point is achieved.

Please explain in more detail how you determine saturation.

Page 15: A semi-structured interview will be used in this study with the help of an interview guide. The interview guide is prepared with the guidance of a few qualitative experts and previous literatures.

Please provide the interview guide as a table or supplementary table. It will be quite key to see how you are asking these questions, given you are a priori selected those participants that scored “poor practice”. Further to this point, there needs to be more information on how you are selecting the parents for the interviews. Are they based on one set of scores or a combination of several scores?

Page 15: There are several areas where you mention examining for face and content validity, there needs to be much more information on how you plan to do this. There should also be suggestions or plans for this if the questions require revision.

Page 15: Please describe more on how the data from the “pilot” will be used. What possible outcomes could occur here? Would you change the questions? Potentially change your sampling frame?

Page 17: …First during the first phase where the quantitative results help plan the follow-up qualitative data collection and the second integration when the results of the quantitative study are connected with the results of the qualitative study to derive a conclusion whether the qualitative results can explain the quantitative results.

Please explain this in a different way – do you mean that your quantitative results will inform your qualitative data collection? And then you will use the results of both to provide a more comprehensive discussion of the results? It is unclear what you mean by follow up qualitative data collection, you are collecting these data twice? Also, please explain how you will ensure the validity of the data collected from the interviews.

Discussion and conclusion: The authors should describe any challenges that may be presented in doing this work and how they plan to mitigate these challenges.

Reviewer #3: This is a study protocol of a mixed method study. I suggest to review the information on the discussions so that this study protocol is reproducible and can help others to conduct the study of same interest.

7. PLOS authors have the option to publish the peer review history of their article (what does this mean?). If published, this will include your full peer review and any attached files.

Reviewer #1: No

Reviewer #2: No

Reviewer #3: No

---

## [Author Response · Author response to Decision Letter 0]

8 May 2022

I responded all reviewers' comment in a separate file . Thank you

---

## [Decision Letter · Decision Letter 1]

15 Dec 2022

PONE-D-20-40899R1Determinants of non-adherence to home injury prevention practice among parents of under-five children in North Seberang Perai district, Penang: A mixed methods study protocolPLOS ONE

Dear Dr. Kadir Shahar,

Thank you for submitting your manuscript to PLOS ONE. After careful consideration, we feel that it has merit but does not fully meet PLOS ONE’s publication criteria as it currently stands. Therefore, we invite you to submit a revised version of the manuscript that addresses the points raised during the review process.

ACADEMIC EDITOR: Please take attention for the comments raised by reviewer 2. Notwithstanding, the manuscript needs additional work to clarify some statements and to ensure clarity. So, I would also encourage you to send it for a professional supply of editing services for English usage.

We look forward to receiving your revised manuscript.

Kind regards,

Thiago Machado Ardenghi

Academic Editor

PLOS ONE

Journal Requirements:

Additional Editor Comments:

Please take attention for the comments raised by reviewer 2. Notwithstanding, the manuscript needs additional work to clarify some statements and to ensure clarity. So, I would also encourage you to send it for a professional supply of editing services for English usage.

Reviewers' comments:

Reviewer's Responses to Questions

**Comments to the Author**

1. Does the manuscript provide a valid rationale for the proposed study, with clearly identified and justified research questions?

Reviewer #1: Yes

Reviewer #2: Yes

2. Is the protocol technically sound and planned in a manner that will lead to a meaningful outcome and allow testing the stated hypotheses?

Reviewer #1: Yes

Reviewer #2: Yes

3. Is the methodology feasible and described in sufficient detail to allow the work to be replicable?

Reviewer #1: Yes

Reviewer #2: Yes

4. Have the authors described where all data underlying the findings will be made available when the study is complete?

Reviewer #1: Yes

Reviewer #2: No

5. Is the manuscript presented in an intelligible fashion and written in standard English?

Reviewer #1: Yes

Reviewer #2: No

6. Review Comments to the Author

You may also provide optional suggestions and comments to authors that they might find helpful in planning their study.

Reviewer #1: Few comments regarding writing typo error in the text such as

1) pg 12 bottom line - repetitive of word index

Author needs to get final proofreading for english grammatical order.

Some of the reference were not written correctly following format ie. the WHO report in PDF from website, must have URL, date accessed

Reviewer #2: Please see attached word document for items that still require attention from the author team. I still do not think that the manuscript is presented in a language that is clear, correct and unambiguous. There is also a response to the reviewers that is not clear and requires revision.

7. PLOS authors have the option to publish the peer review history of their article (what does this mean?). If published, this will include your full peer review and any attached files.

Reviewer #1: No

Reviewer #2: No

---

## [Author Response · Author response to Decision Letter 1]

29 Jan 2023

Thank you for the constructive comments. The repetition has been removed. We have done final proofreading for English grammatical order of the manuscript.

Thank you for the suggestion for reference. However, the suggested study was an interventional study that introduce interventions such as education and training to improve parental health belief constructs and motivation to conduct preventive measures. It cannot be applied to this study as we measure how parental prevention practice influenced by parental, child and environmental factors. This study has very limited references.

The manuscript has been proofread.

 Although the relationship between child injury and its prevention has been established by previous studies, subsequent researchers tend to directly relates injury (outcome) to many variables or associated factors and exclude preventive measures (that may have or have not been taken) by the parent/guardian. Thus, this study is trying to fill that gap in terms of home injury among the Malaysian population.

The quantitative study will be using Cross-sectional design. This is used as we are looking into the association parental home injury prevention practice with many variables which is suitable to be studied using questionnaire at one point of time. The following qualitative study plans to explains the findings from the quantitative study, which can only be answered by the same respondents. This will be achieved by using in-depth interview that is suitable for a sensitive and private issue such as parenting and child injury where parents can be embarassed to discuss in a group (FGD).

Selection bias that may occur due to the study location is the sample may consist more of mothers home-makers, lower income and lower education as these clinics' clients are largely made of these demographic groups.

We thank the reviewer for the valuable suggestion regarding our discussion and conclusion. As suggested by the reviewer, we have added our challenges in doing this research and their solution in our manuscript and were highlighted in yellow.

---

## [Decision Letter · Decision Letter 2]

1 Mar 2023

Determinants of non-adherence to home injury prevention practice among parents of under-five children in North Seberang Perai district, Penang: A mixed methods study protocol

PONE-D-20-40899R2

Dear Dr. Kadir Shahar,

We’re pleased to inform you that your manuscript has been judged scientifically suitable for publication and will be formally accepted for publication once it meets all outstanding technical requirements.

Kind regards,

Thiago Machado Ardenghi

Academic Editor

PLOS ONE

Additional Editor Comments (optional):

Reviewers' comments:

Reviewer's Responses to Questions

**Comments to the Author**

1. Does the manuscript provide a valid rationale for the proposed study, with clearly identified and justified research questions?

Reviewer #1: Yes

2. Is the protocol technically sound and planned in a manner that will lead to a meaningful outcome and allow testing the stated hypotheses?

Reviewer #1: Yes

3. Is the methodology feasible and described in sufficient detail to allow the work to be replicable?

Reviewer #1: Yes

4. Have the authors described where all data underlying the findings will be made available when the study is complete?

Reviewer #1: Yes

5. Is the manuscript presented in an intelligible fashion and written in standard English?

Reviewer #1: Yes

6. Review Comments to the Author

You may also provide optional suggestions and comments to authors that they might find helpful in planning their study.

Reviewer #1: Dear Authors,

Generally, authors have improved the article by addressing all the important comments given by the reviewers. The limitation, justification also have been addressed and answered accordingly by the authors. However, I have a few technical errors that I noticed that may need to be corrected before this article get to be published.

At pg 9, for the last paragraph, comma (,) is missing after sentences" substandard safety measures in their low-income house,

Under section Data Analysis, symbol for chi square test need to be corrected. the X2, the 2 is supposed to be written as superscript form.

Under section Data Collection Method (for Phase II Qualitative study), at paragraph 5, which talks about Pilot study of Qualitative data. The last sentence that stated 'Moreover, by getting the result from Pilot test, items that show Cronbach Alpha value below 0.7, will be deleted from the questionnaire to achieve better internal consistency". I think CA value is not relevant here cause the paragraph talks about Qualitative study.

As a conclusion, I think this paper has fulfilled the reviewer's previous comments and shall be published once the technical/typo error are addressed.

7. PLOS authors have the option to publish the peer review history of their article (what does this mean?). If published, this will include your full peer review and any attached files.

Reviewer #1: No

---

## [Editor Report · Acceptance letter]

7 Mar 2023

PONE-D-20-40899R2 

Determinants of non-adherence to home injury prevention practice among parents of under-five children in North Seberang Perai district, Penang: A mixed-methods study protocol 

Dear Dr. Kadir Shahar:

I'm pleased to inform you that your manuscript has been deemed suitable for publication in PLOS ONE. Congratulations! Your manuscript is now with our production department. 

Kind regards, 

on behalf of

Dr. Thiago Machado Ardenghi 

Academic Editor

PLOS ONE